# Adaptation of the Pet Bereavement Questionnaire for European Portuguese Speakers

**DOI:** 10.3390/ijerph20010534

**Published:** 2022-12-28

**Authors:** Isabel Silva, Glória Jólluskin, Estela Vilhena, Allison Byrne

**Affiliations:** 1I3ID, Human and Social Sciences Faculty, Universidade Fernando Pessoa, 4249-004 Porto, Portugal; 22Ai, School of Technology, Instituto Politécnico do Cávado e do Ave, 4750-810 Barcelos, Portugal

**Keywords:** pet, bereavement, questionnaire

## Abstract

The loss or death of a beloved pet creates a grief reaction comparable to that of the loss of a family member and may lead to the development of symptoms consistent with a diagnosis of persistent complicated grief disorder. Nevertheless, society does not always recognize it as a significant loss, which may contribute to bereaved owners feeling isolated and ashamed when coping with it, as well as not resorting to mental health professionals when necessary. The development of instruments to assess these reactions may contribute to improving the understanding of this suffering. This study aimed to adapt the Pet Bereavement Questionnaire for European Portuguese speakers. A non-probabilistic convenience sample of 169 adults who had a pet that died answered a battery of questionnaires, which included the Portuguese version of the Pet Bereavement Questionnaire. This version resulted from a consensus translation prepared by two translators and subsequently subjected to a cognitive debriefing. The Portuguese version of the instrument demonstrated good reliability (good internal consistency for the total questionnaire and for its subscales) and good external validity (negative correlation with well-being measures and positive correlation with psychopathology measures), as well as reasonable internal validity and sensitivity.

## 1. Introduction

Over the past four decades, in the field of human-animal interactions, there has been a growing interest in bereavement related to pet death. There is no doubt as to the significant role that pets play in our lives. Twenty years ago, Podrazik et al. [1] pointed out the increasing difficulty in defining the concept of “family” and that any such definition should take into consideration the significant role played by animals in the dynamics of family systems. More recently, Doré et al. [2] highlighted two ways of living with pets. In the first, the pet was perceived as an object or property. In the second, the pet was perceived as a subject, a friend or companion, as a “family member”. It is from this second perspective that bereavement processes associated to pet loss can be studied.

Podrazik et al. [1] drew attention to the fact that, at some point, pet owners have to face the fact that many of their animals will predecease them and that they will have to deal with their deaths, or even with the responsibility of deciding whether or not to euthanize them.

One may wonder if grief over pet death differs from grief over the death of a human loved one. Research has long suggested that the loss or death of a beloved pet creates a grief reaction comparable to that of the loss of a human family member [3]. Some owners described the relationship with their pets as fulfilling an intimate social human role (such as best friend or child), anthropomorphizing them [4]. Hence, the death of a pet could be painful and cause reactions such as anxiety, stress, shame, ambiguous grief, complicated grief and even trauma [5]. These reactions could lead to preoccupation with thoughts of the pet, poor concentration, avoidance behaviour, loss of identity and self-reproach [5]. Bereaved pet owners also reported disruption in daily living and social activities [6]. However, very recently, Lee [7] went a step further and conducted a study aimed at validating the construct of persistent complicated grief disorder for bereaved pet owners. He concluded that pet grief symptoms are consistent with the symptom classification proposed by The Diagnostic and Statistical Manual of Mental Disorders, Fifth Edition (DSM-5) and that they occur in both men and women. Individuals who have lost a pet and who have long-term grief disorder show more symptoms of depression, loneliness, difficulty sleeping, show more negative religious coping, and more coping through alcohol/drug use than do other individuals who have lost their pets and do not have this disorder. 

This finding led Lee [7] to support the application of the DSM-5 model of persistent prolonged grief disorder to owners whose pets had died. Lee observed that 9.4% of the participants in his study presented criteria for diagnosis of this disorder, noting that this value was really close to that found by Lundorff et al. [8] of (9.8%) in a meta-analysis conducted to estimate the prevalence of symptoms of complicated grief disorder in adults who had lost somebody close to them. These values were higher than those presented by the American Psychiatric Association [9] in the DSM-5 for persistent complicated grief disorder (2.4–4.8%), but, even if they were similar, pet grief can no longer be underestimated. 

Society does not always recognize the significant loss associated with the death of a pet. This lack of loss recognition can contribute to bereaved owners feeling isolated and ashamed when dealing with the loss of their companion [10]. Peseschkian [11] stressed that social and religious norms regarding grieving behaviour can encourage or inhibit the grieving process.

Thus, mental health professionals should be aware of what Hughes et al. [5] and Hess-Holden, et al. [12] referred to as “unprivileged grief” and provide support to bereaved pet owners. Veterinarians, who deal very closely with suffering owners, must also be sensitive to owners’ psychosocial reactions and be capable of directing them to specialized support if needed.

Denial by veterinarians of the grief caused by animal death does not help pet owners to overcome it and may even exacerbate the extent of the distress they feel. In particular, veterinarians’ relational and communication skills and their approach in providing treatment to pets may alleviate or aggravate the owners’ grief [13].

Research reveals the importance of a number of variables in managing pet loss, notably the sensitivity of the veterinarian, having a veterinarian who helps owners make informed and conscious decisions about their pet, and choosing the right time to perform euthanasia. However, these factors may prove insufficient [10], in which case it is important that veterinarians are prepared to offer psychological first aid, and to guide owners towards psychologically appropriate professional help where required.

Dow et al. [14] pointed out that most veterinarians did not refer any grieving clients to a psychologist or other mental health professional. Dow et al. [14] further noted that a significant proportion of veterinarians felt that their own mental health was affected when dealing with clients suffering from the loss of a pet.

In this context, it would seem to be extremely important to develop bereavement assessment tools, aimed at those who have experienced the loss of a pet, which allows for a greater understanding of their reactions, as well as assessing the need for their referral to mental health care.

The instruments available in this field are scarce. In a first analysis we found what seemed to be several instruments, but a closer examination showed the existence of a single one: Pet Bereavement Questionnaire (PBQ) [15], since Mourning Dog Questionnaire (MDQ) [16] is, in fact, a battery of instruments in which grief is assessed through an Italian version of the PBQ. The same is true for the pet bereavement assessment battery created by Testoni et al. [13] which assesses bereavement with an Italian adaptation of the PBQ. Also, the Australian assessment battery of Spain et al. [17] included the PBQ and the Loss of Social Support subscale of the Grief Experience Questionnaire.

Thus, the PBQ developed by Hunt and Padilla in 2006 [15], undoubtedly stands out, with its goal of creating a brief and valid instrument for use in studies on the psychological impact of losing a pet. The original version of this instrument consists of a Likert-type scale comprising 16 items, organized into three subscales: Anger, Grief and Guilt, for which four response options are offered, ranging from “strongly disagree” to “strongly agree”. It shows adequate reliability and validity and has potential for research contributing to the identification of individuals at elevated risk (including in veterinary hospitals) and to the evaluation of the results of interventions [15].

It is essential to have valid and reliable instruments that are appropriate to the specificities of each country. However, it is also important to have instruments that are comparable across diverse cultural realities. Only by having rigorous assessment measures that can be administered in different realities, can we have a better understanding of the psychological needs of people who have to deal with the death of their pets. Furthermore, the use of scientifically validated measuring tools is an essential condition for the development of knowledge that will allow us to think about the training of mental health professionals and veterinarians, equipping them with specific skills that will render them more competent to deal with bereaved owners. 

Thus, the creation of assessment instruments for isolated national or regional studies is not enough. It is essential that there be a joint effort by researchers from different nationalities, adapting for different countries those measures whose psychometric qualities have already been empirically demonstrated. The present study aimed to adapt and validate PBQ to the European Portuguese language and culture, evaluating its psychometric qualities.

## 2. Materials and Methods

### 2.1. Participants

A non-probabilistic convenience sample was studied, consisting of 169 participants who had a pet that died: 89.3% (n = 151) female; 89.3% (n = 151) living in urban areas; aged between 18 and 82 years (M = 34.72; SD = 14.07); with a level of education ranging from years 7-9 of basic education to doctorate, with 78.1% (n = 132) attending higher education level. All participants, at the time of their participation in the study, had at least one pet, with this number ranging between 1 and 25, and on average they had 3 pets (SD = 3.09): 68.6% (n = 116) dog, 50.9% (n = 86) cat, 7.1% bird (n = 12), 10.1% fish (n = 17), 6.5% (n = 11) guinea pig, 3% (n = 5) reptile, and 9.5% (n = 16) other animals.

### 2.2. Instruments

Participants responded to the following instruments:

(a) Socio-demographic and pet characterization questionnaire, developed specifically for the present study with the objective of collecting data on gender, age, education, and, if they currently have pets, what pets these are and about participants’ general health perception.

(b) Mental Health Continuum-Short Form (MHC-SF) Portuguese version [18]: consisting of 14 items that aim to assess well-being domains (emotional, social and psychological), offering a Likert scale with six response options. The Portuguese version showed good validity, good sensitivity and good reliability [18]. The higher the score obtained, the higher the individual’s well-being. In this study, the instrument presented a Cronbach’s alpha of 0.95

(c) Warwick-Edinburgh Mental Well-Being Scale (WEMWBS) Portuguese version: this scale is being adapted to Portugal by Fonte et al. (version under study) [19]. This instrument assesses mental health, focusing only on its positive aspects. It consists of 14 items organized in a single factor, with good psychometric qualities. The higher the score obtained, the greater the mental well-being. In this study, the instrument presented a Cronbach’s alpha of 0.87.

(d) Life Satisfaction Scale Portuguese version (made available by Diener—https://internal.psychology.illinois.edu/~ediener/SWLS.html): a scale that evaluates life satisfaction. It includes five items that constitute statements to which the individual answers on a Likert-type scale of seven positions, ranging from “totally disagree” to “totally agree”. The higher the score obtained, the higher the life satisfaction reported. In the present study, this instrument presented a Cronbach’s alpha of 0.88. 

(e) Anxiety, Depression and Stress Scale (SADS) Portuguese version [20]: an instrument that aims to assess symptoms associated with anxiety, depression and stress, consisting of 21 items grouped into three subscales of seven items each. All items are assessed using a 4-point Likert-type response scale ranging from “it did not apply to me at all” to “it applied to me most of the time”. Higher values correspond to more negative affective states. The Portuguese version of the instrument revealed good psychometric qualities [16]. In the present study, it presented a Cronbach’s alpha of 0.96.

(f) Fear of Happiness Scale Portuguese version (FHS) [21]: this instrument assesses fear of being happy and is composed of five items, organized into a single factor. All items are assessed using a concordance scale with seven response options, ranging from “Strongly disagree” to “Strongly agree”. The Portuguese version revealed good reliability and validity. In the present study, it presented a Cronbach’s alpha of 0.91. Higher values correspond to higher levels of fear of happiness.

(g) Pet Bereavement Questionnaire (PBQ) Portuguese version: the original version of this instrument was developed by Hunt and Padilla in 2006 [15] in the United States of America. The Portuguese version under study consists of 16 items, organized into three subscales: Guilt (items 6, 8, 9 and 16), Grief (items 2, 3, 5, 7, 10, 12 and 15), and Anger (items 1, 4, 11, 13 and 14). A Likert-type response scale is offered, with four response options ranging from strongly disagree (0) to strongly agree (3). The total score is calculated from the sum of its items. Thus, for the total scale the minimum value is 0 and the maximum is 48. The score of each subscale is calculated through the sum of its items, dividing this value by the total number of items that constitute the subscale. Thus, for each subscale, the minimum value will be 0 and the maximum value 3. 

### 2.3. Procedure

Permission was requested from the authors of the original version of the PBQ [15] to proceed with its adaptation for the Portuguese language and culture. The translation of the PBQ into Portuguese was carried out by two translators fluent in Portuguese and English and with scientific knowledge in the field studied. These two independent versions were later compared, discrepancies were explored, and alternatives and possible changes were discussed in order to improve the quality of the translation, ensuring that it was linguistically correct and equivalent in terms of content, to the original version. This joint work resulted in a single consensus version, subjected to a cognitive debriefing with a small group of individuals. This debriefing aimed to detect possible problems of ambiguity, difficulty in understanding or cultural barriers, allowing for the removal of any inappropriate or offensive language. 

Following approval from the Ethics Committee of Fernando Pessoa University, an invitation to participate in the study was made. Different approaches were adopted: (a) a message was sent through institutional mailing lists; (b) an invitation was sent through social networks (Facebook); (c) an invitation directly invited respondents to cooperate in answering the questionnaires; and/or (d) an invitation was forwarded to other potential participants, generating a non-random, “snowball” sample.

Inclusion criteria for participants were as follows: (a) to be 18 years of age or older; (b) to be able to give free and informed consent; (c) to have experienced the loss of a pet with participants answering yes to the question, “Have you suffered the loss of one of your pets?” Participants completed the questionnaires after giving their informed consent and were offered no incentives to participate in the study.

The questionnaires were computerized so that they could be administered electronically and were self-completed online by the respondents. No personal identification data was collected from the participants to ensure anonymity and data confidentiality. 

Since the topic of grief may be perceived as sensitive, participants were informed that, if they felt disturbed after completing the questionnaire, their general practitioner could provide them with follow-up care.

A sample size must be determined to achieve adequate questionnaire precision and to have enough power to reject a false null hypothesis. A literature review by Anthoine et al. [22] concluded that recommendations ranged from 2 to 20 subjects per item, with an absolute minimum of 100 to 250 subjects. In the present study the following criteria were defined a priori to assure precision: assessing at least 10 participants for each scale item, i.e., the ratio of respondents to items of 10:1. Furthermore, as there were no missing answers to the items, appropriate handling of incomplete outcome data was not necessary.

### 2.4. Statistical Analysis

Descriptive statistics (minimum, maximum, mean, standard deviation and median values, asymmetry and kurtosis) were calculated to describe the responses to the items. Exploratory factor analysis (EFA) was used to analyse the factor structure of the questionnaire and Cronbach’s alpha was used to test internal consistency of the items. Pearson’s correlation, corrected for overlapping, was used to analyse item-total and item-subscale correlation with the intention of testing internal validity. Pearson’s correlation coefficients were calculated to analyse correlation between the Portuguese version of the PBQ, measures of well-being and psychopathology, age, and the number of pets each individual had. We also used the Mann-Whitney U test to analyse differences in gender and area of residence, as well as the One-Way-ANOVA test to check for differences between participants with different levels of education with regard to the scores obtained in the PBQ and the subscales of this instrument. 

Significance level was set at 0.05 throughout the analyses. Statistical procedures were done using SPSS Statistics 26.0 (IBM, Porto, Portugal). 

To test how well the measured variables represent the factor structure of the PBQ, confirmatory factor analysis (CFA) was used. Three continuous latent variables were regressed (Guilt, Grief and Anger). Maximum likelihood estimation was used to assess the fit of the model; however, this method assumes that the data must have a normal multivariate distribution [23], so the more robust chi-square statistic, the Satorra-Bentler scaled statistic ($SB_{\chi^{2}}$) [24], was used. This test corrects for the non-normality of the data distribution and produces more accurate results [25]. The goodness of fit for each factor structure was evaluated using several descriptive criteria: the ratio between Chi Square and degrees of freedom ($\chi^{2}/df)$, the Normal Fit Index (NFI), Comparative Fit Index (CFI) and the Root Mean Square Error of Approximation (RMSEA) with its 90% confidence interval (RMSEA 90% CI). As recommended by Brown [26], the model was considered to have “adequate fit” if the RMSEA was less than 0.08 and the CFI was greater than 0.9; “good fit” was indicated by an RMSEA less than 0.05, a CFI greater than 0.95 and $\chi^{2}/df$ values less than 3. For each variable, the magnitudes of factor loadings were considered. Variables with a factor loading of 0.3 or greater [26] were considered representative of the construct being measured in each domain. The CFA was performed with the use of EQS 6.1 software [27].

## 3. Results

A descriptive analysis of the values obtained in each item, in the 3 subscales and in the total scale was carried out to analyse the sensitivity of the Portuguese version of the PBQ (Table 1).

The values found in the items varied between the extreme points of the scale (0 and 3) and the mean and median values were practically overlapping. The values of asymmetry and kurtosis were mostly close to zero, not exceeding unity for most items (the exceptions being items 1, 4, 11, 13 and 14). There were no missing answers to the items of the Portuguese version of the PBQ.

Kaiser-Meyer-Olkin (KMO = 0.88) and Barlett’s test of sphericity ($\chi_{120}^{2}$ = 1517.491, *p* < 0.0001) were calculated. These results showed a good suitability of the data for performing factor analysis, according to the criteria advocated by Maroco [28]. 

The Portuguese version of the PBQ has a Cronbach’s alpha of 0.91, and the internal consistency does not benefit from the elimination of any item from the questionnaire. Its subscales also showed good internal consistency: Guilt α = 0.88; Grief α = 0.90; Anger α = 0.70.

Given the clinical relevance of the 3 subscales of the PBQ, we performed a correlation analysis, corrected for overlap, item-total and item-subscales with the purpose of further assessing their internal validity (Table 2).

The item-total correlations, corrected for overlap, showed a moderate to strong association. The item-subscale correlations, with equal correction, revealed that the item-subscale correlation to which it belongs is moderate to high and higher than the correlation with the other 2 subscales for 11 of the 16 items that make up the PBQ.

The correlation between the 3 subscales was statistically significant, positive, moderate to high, and the correlation between these subscales and the Total PBQ was high (Table 3).

In order to test the external validity of the instrument, we analysed the association between the Total PBQ, its three subscales and the remaining administered measures of psychological well-being and distress (Table 3).

The total PBQ and its three subscales showed a statistically significant negative and weak correlation with psychological well-being, emotional well-being, social well-being, mental well-being, life satisfaction and general health perception, and a statistically significant, positive and weak correlation with anxiety, depression and stress. We highlight the fact that the subscales Guilt and Grief were statistically significant, and positively and weakly correlated with the fear of being happy and that the total PBQ and the Anger subscale were statistically significant, and positively and moderately correlated with this fear.

The Mann-Whitney U test was used to compare male and female participants, and it was found that there were no statistically significant differences regarding total PBQ, nor regarding the three subscales of this instrument (*p* > 0.05).

The Pearson’s correlation analysis showed that there was no statistically significant correlation between age and Grief and Anger subscales, but revealed that there was statistically significant, negative and weak correlation between age, total PBQ (r = −0.19; *p* < 0.05) and Guilt subscale (r = −0.29; *p* < 0.0001). To clarify the explanatory hypothesis advanced by Hunt and Padilla [15] for this latter association that younger people would be those who present higher levels of depressive symptoms and, therefore, also more grief symptoms, we analysed the correlation between participants’ age and depressive symptoms. The analysis revealed that there was a statistically significant, negative and weak correlation between age and these symptoms (r = −0.34; *p* < 0.0001).

The One-Way ANOVA test revealed that there were statistically significant differences between participants with different educational levels on the total PBQ (F(4; 160) = 2.687; *p* < 0.05) and Anger subscale (F(4; 160) = 2.633; *p* < 0.05), although not for the remaining subscales (*p* > 0.05). The higher the level of education, the lower the value obtained in the total PBQ (years 7–9 of basic education M = 30.50; SD = 6.36; secondary education M = 20.18; SD = 10.54; undergraduate degree M = 18.32; SD = 9.47; master’s degree M = 15.49; SD = 9.53; doctorate (M = 14.32, SD = 10.71) and in the Anger subscale (years 7–9 of basic education M = 1.6, SD = 0.28; secondary education M = 0.67, SD = 0.60; undergraduate degree M = 0.60, SD = 0.60; master’s degree M = 0.48, SD = 0.54; doctorate M = 0.43, SD = 0.48).

The Mann-Whitney U-test was calculated and no statistically significant differences were found between individuals living in rural and urban environments with regard to the total score obtained in the PBQ, nor with regard to the 3 subscales of the PBQ (*p* > 0.05). However, it should be noted that participants living in rural areas had a lower level of education (34.9% with education levels equal to or lower than high school) than those living in urban areas (17.5% with education equal to or lower than high school) (K-S = 1.601; *p* < 0.05).

Finally, we analysed the correlation between the number of pets the individuals had at the time of participation in the study and the grief symptoms presented, having verified that the higher the number of pets owned, the higher the score obtained in the PBQ (r = 0.23; *p* < 0.01) and in the subscales Grief (r = 0.19; *p* < 0.05), Guilt (r = 0.19; *p* < 0.05) and Anger (r = 0.20; *p* < 0.01).

The results of the CFA revealed that the 16-item, 3-factor model of the Portuguese version of the PBQ (latent variables: Guilt, Grief and Anger) provided a good fit model for the data (see Figure 1). All items significantly loaded their hypothesized factors and good fit indices, $S-B_{\chi^{2}}$ = 109.8; df = 70; *p* < 0.001; NFI = 0.92; CFI = 0.97; RMSEA = 0.059; 90%CI = [0.036, 0.079]. Standardized factor loadings greater than 0.3 were also observed for all items.

## 4. Discussion

With regard to the distribution of the values of the Portuguese version of the PBQ, the range of responses is very similar to that found in the original version (40 and 42, respectively), although the mean and median values for the total scale and the three subscales in the Portuguese version are lower than those found in the study by Hunt and Padilla [15] in the United States of America and by Uccheddu et al. [16] in Italy. 

The analysis of the distribution and the asymmetry and kurtosis values for each item, for the three subscales and for the total scale seem to indicate that, in general, the questionnaire has a normal distribution according to the criteria proposed by Maroco [28], suggesting that it is a sensitive instrument, i.e., that the results appear distributed, differentiating the subjects in their symptomatic levels. In items in which asymmetry and kurtosis were higher than one, i.e., responses were not normally distributed, all response options offered, from “Disagree Strongly” to “Agree Strongly”, had answers. Moreover, we note that these items belong mainly to the Anger subscale, assessing anger (toward people who may have contributed to the animal’s death and toward friends and family for not having helped more) and trauma (nightmares, feeling tormented by memories of the pet), domains in which, from a clinical point of view, one would not anticipate the occurrence of a normal distribution. These results are not in line with those presented by Hunt and Padilla [15], who had found a normal distribution in Guilt and Anger subscales. 

The Portuguese version of the PBQ revealed high reliability. The Grief and Guilt subscales also revealed high reliability and the Anger subscale presented reasonable reliability. The internal consistency for the Portuguese version of the PBQ was higher overall than in the original version (α = 0.87) [15]. It was also higher than in the Turkish version (α = 0.87) [29]. Ribeiro [30] considers that a good internal consistency should exceed a Cronbach’s alpha of 0.80 whereas Almeida and Freire [31] consider values between 0.70 and 0.80 as representing a respectable internal consistency, and between 0.80 and 0.90 very good internal consistency. Nevertheless, these authors warned that, when a value is higher than 0.90, a reduction in the number of items should be considered, as it may indicate too much homogeneity [30,31]. Although the items are not redundant, in future studies in which the PBQ is tested with clinical populations, it will be important to analyse whether items can be removed from the instrument, making it even shorter, without however, losing relevant content. 

The 3-factor structure, found in the original version of the PBQ [15] and in the Turkish version [29], was verified in the Portuguese version through Confirmatory Factor Analysis. Item-subscale analysis seems to be, in general, consistent with an acceptable internal validity when considering the 3 factors originally proposed, even though, for some items, the theoretical independence between the domains assessed - Guilt, Grief and Anger - may be questioned. The correlation between the 3 subscales and between these and the total PBQ also seems to support the internal validity of the instrument. These correlations are higher than those found by Hunt and Padilla [15] in the original version and by Uccheddu et al. [16] in the Italian version.

The higher the score obtained in the PBQ and its subscales, the lower the psychological well-being, emotional well-being, social well-being, mental well-being, life satisfaction and general perception of health of the person whose pet died. Likewise, the higher the score obtained in the PBQ and its subscales the higher the symptoms of depression, anxiety and stress, as well as their fear of being happy. These results support the external validity of the Portuguese version of the PBQ and match those found by Hunt and Padilla [15], who revealed the existence of a significant and strong association between symptoms of depression (assessed through the Beck Depression Inventory) and the original version of the PBQ. 

Men and women did not differ in the total score of the PBQ, nor in the scores obtained in its three subscales. The number of men assessed is small, so this aspect should be further investigated in future studies. In any case, it is in agreement with the results of Hunt and Padilla [15] (who also studied a sample with a small number of male participants) and may mean that symptoms of long-term grief disorder occur in both men and women, as demonstrated by Lee’s study [7]. Nevertheless, it is probable that cultural background plays a role in the pet bereavement process in women and men. In the Turkish population, PBQ showed that women reported more symptoms of bereavement than men [29]. 

The higher the age, the lower the grief symptoms for the death of a pet and the lower the guilt felt, although no significant association was found when considering the Grief and Anger subscales. Uccheddu et al. [16] found a negative association between age, total PBQ and Anger and Guilt subscales, but not with the Grief subscale. Hunt and Padilla [15] also found a negative and weak correlation between age and Guilt. They posited the explanation that, with age, individuals begin to understand that loss is inevitable and therefore feel less responsible for their own role in this process. Also, being more likely to have previously experienced more pet loss, they have lower levels of depression due to the most recent pet death in their lives than is the case for younger individuals [15]. Moreover, the prevalence of extreme depressive symptoms may decline with age, even though the mean number of depressive symptoms in the population may increase [32].

Although one of the limitations of the present study was the fact that it did not control whether individuals had previously suffered the loss of other pets or significant others, its results reveal that the older the individuals, the lower the intensity of depressive symptoms they present, which seems to support this hypothesis advocated by Hunt and Padilla [15]. The present study carried out with the Portuguese population, the study carried out with the American population [15] and the study carried out with the Italian population [16] point out that Grief is also present regardless of the age of the bereaved owner.

Individuals living in rural and urban settings did not differ when it comes to grief symptoms (total scale score, Guilt, Grief, and Anger subscales). Nevertheless, the lower their education level, the more grief symptoms individuals showed (total scale and Anger subscale). On the other hand, data revealed that participants living in rural areas have lower levels of education. Uccheddu et al. [16] also found a statistically significant negative and weak relationship between levels of education and scores on the total PBQ and the Grief and Anger subscales. 

These results are an important contribution towards deconstructing a false preconception among the general population that, in rural areas, animals are still seen essentially as an object and not as a subject (as these concepts are understood by Doré et al. [2]). According to this preconception, people living in rural areas would be less likely to develop grief processes in reaction to the death of their animals. This prejudice might lead the community in general and health professionals, in particular, to underestimate reactions to the death of a pet in people from these areas. Our results suggest that in rural areas there is a clear differentiation between pets, livestock and pests.

In addition, as the population living in rural areas has lower academic qualifications, they may have lower health literacy. The study on the state of health literacy in Portugal conducted by Espanha, Ávila, and Mendes [33], revealed that more than 60% of respondents with an education level of up to elementary school have problematic or even inadequate levels of health literacy. More recently, the study by Carneiro, Silva, and Jólluskin [34] revealed that the lower the level of education, the lower the level of health literacy. Thus, people with lower educational attainment may find it more difficult to search for mental health information related to pet death and to seek health services, as well as to communicate with health professionals, about how they are coping with this loss.

The more pets a person currently has, the higher the level of grief symptoms, both in terms of total score and in each of the individual domains assessed - Grief, Guilt and Anger. When the death of an animal occurs, there is often an absence of social recognition or validation that the bereaved owner has the right to grieve or claim social support, with grief being a process devoid of privilege [35]. Grief over the death of a pet may not be fully recognized or validated as a significant loss, resulting in what may be termed an “unrecognized grief”, a grief experienced in isolation and without support. This lack of recognition makes it common for those around the grieving owners to encourage them to get a new animal soon after the death of their beloved pet. However, the results of this study show owning pets does not necessarily mitigate the intensity of the loss and grief symptoms.

One of the limitations of the study stems from the procedure adopted for data collection. By administering the assessment battery using only the self-completion of an electronic form, all individuals with difficulty in completing questionnaires autonomously and needing support to answer were excluded. Also excluded were those without Internet access. Future studies should consider combining other data collection procedures to evaluate more representative samples of the population. Despite this limitation, there is an advantage of responding via electronic form because it frees respondents from any embarrassment that might exist in recognising how they feel about the death of their pets.

Other limitations relate to the fact that test-retest over a short period of time was not performed, which would have allowed for the exploration of temporal stability of the instrument, nor was the time elapsed since the death of the animal or the circumstances of the animal’s death controlled, which could have allowed other analyses of the sensitivity of the PBQ.

The negative association with well-being-related variables and positive association with symptoms of psychopathology reinforce the importance of using measures such as PBQ for identifying an individual’s referral need for support. Grief symptoms have been shown to be important in an individual’s global assessment of their own lives, their well-being and their health. Grief symptoms also proved to be associated with psychological distress; namely, anxiety, depression, stress and the fear of being happy. It should not be forgotten that general health perceptions are significantly and independently associated with specific health problems, use of health services, changes in physical status, recovery from illness episodes, and is a strong predictor of mortality in the general population [36]. These results should, therefore, not be underestimated; grief over pet loss can have a serious impact on the lives and health of bereaved owners.

We believe that it is important to conduct studies with the purpose of defining cut-off points to differentiate between individuals who present diagnostic criteria for long-term complicated grief and those who do not present this disorder, so as to enhance the usefulness of the PBQ as a screening tool for possible clinical cases requiring psychological assessment and follow-up by mental health professionals, as well as to test whether the PBQ may be appropriate to assess grief in situations in which the loss of a loved animal is not due to its death (for example, in the case of pet disappearance).

## 5. Conclusions

In conclusion, the Portuguese version of the PBQ is a self-report instrument that has proven to be reliable, valid and sensitive, was well accepted by the respondents, and is a quick-response questionnaire that can be used regardless of the individuals’ area of residence and gender.

## Figures and Tables

**Figure 1 ijerph-20-00534-f001:**
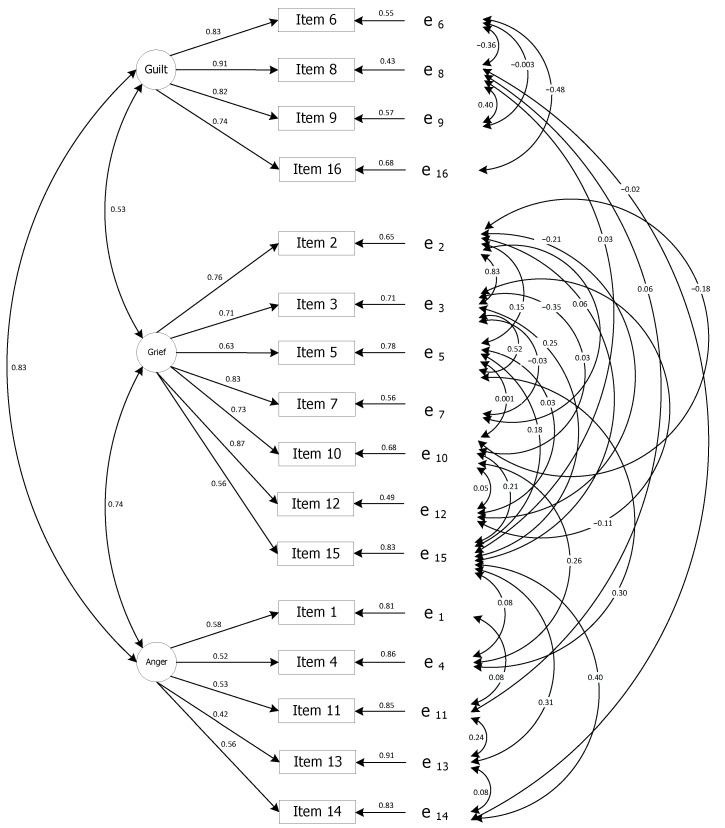
Factor loadings of confirmatory factor analysis for the Portuguese version of the PBQ (Standardized Parameters Estimates) and residual errors (e_1_ to e_16_).

**Table 1 ijerph-20-00534-t001:** Descriptive analysis of the values obtained in the PBQ.

	Min	Max	M	Med	SD	Asymmetry	Kurtosis
Item 1	0	3	0.53	0	0.76	1.36	1.26
Item 2	0	3	1.59	2	1.05	−0.16	−1.15
Item 3	0	3	1.58	2	1.04	0.14	−1.13
Item 4	0	3	0.68	0	0.92	1.14	0.21
Item 5	0	3	1.22	1	1.04	0.26	−1.17
Item 6	0	3	1.11	1	0.98	0.37	−0.98
Item 7	0	3	2.04	2	0.86	−0.84	0.35
Item 8	0	3	0.90	1	1.04	0.78	−0.70
Item 9	0	3	0.93	1	1.03	0.68	−0.85
Item 10	0	3	1.47	1	0.97	0.04	−0.95
Item 11	0	3	0.59	0	0.99	1.52	0.96
Item 12	0	3	1.96	2	0.98	−0.79	−0.29
Item 13	0	3	0.34	0	0.69	2.31	5,21
Item 14	0	3	0.71	0	0.88	1.04	0.13
Item 15	0	3	0.81	1	0.89	0.80	−0.29
Item 16	0	3	0.99	1	1.12	0.64	−1.08
Total PBQ	0	40	17.46	17.00	10.09	0.16	−0.67
Guilt	0	3	0.98	0.75	0.89	0.66	−0.59
Grief	0	3	1.52	1.57	0.77	−0.15	−0.55
Anger	0	2.40	0.57	0.40	0.58	0.98	0.31

**Table 2 ijerph-20-00534-t002:** Item-total and item-subscale correlation corrected for overlap.

	Total PBQ	Guilt	Grief	Anger
Item 1	0.51	0.51	0.36	0.51
Item 2	0.63	0.36	0.70	0.47
Item 3	0.66	0.38	0.79	0.43
Item 4	0.56	0.34	0.60	0.37
Item 5	0.59	0.30	0.72	0.42
Item 6	0.65	0.66	0.48	0.56
Item 7	0.63	0.42	0.71	0.40
Item 8	0.69	0.86	0.44	0.62
Item 9	0.66	0.82	0.42	0.59
Item 10	0.63	0.35	0.70	0.50
Item 11	0.47	0.47	0.32	0.37
Item 12	0.69	0.45	0.75	0.48
Item 13	0.44	0.43	0.28	0.51
Item 14	0.55	0.53	0.41	0.48
Item 15	0.68	0.51	0.60	0.60
Item 16	0.55	0.62	0.36	0.52

**Table 3 ijerph-20-00534-t003:** Association between mourning and measures of psychological well-being and distress.

	Total PBQ	Grief	Guilt	Anger
Psychological Well-being	r = −0.24 *	r = −0.17 *	r = −0.25 **	r = −0.23 *
Emotional Well-being	r = −0.32 ***	r = −0.30 ***	r = −0.28 ***	r = −0.20 *
Social Well-being	r = −0.30 ***	r = −0.23 **	r = −0.33 ***	r = −0.21 **
Mental Well-being	r = −0.29 ***	r = −0.22 **	r = −0.30 ***	r = −0.22 **
Life Satisfaction	r = −0.31 ***	r = −0.27 ***	r = −0.23 **	r = −0.30 ***
General Health Perception	r = −0.26 *	r = −0.23 **	r = −0.19 *	r = −0.24 *
Anxiety	r = 0.36 ***	r = 0.31 ***	r = 0.31 ***	r = 0.29 ***
Depression	r = 0.39 ***	r = 0.32 ***	r = 0.38 ***	r = 0.30 ***
Stress	r = 0.39 ***	r = −0.31 ***	r = 0.38 ***	r = 0.31 ***
Fear of Happiness	r = 0.44 ***	r = 0.33 ***	r = 0.37 ***	r = 0.46 ***
Total PBQ	-	r = 0.88 ***	r = 0.81 ***	r = 0.84 ***
Grief	-	-	r = 0.49 ***	r = 0.59 ***
Guilt	-	-	-	r = 0.67 ***
Anger	-	-	-	-

* *p* < 0.05; ** *p* < 0.01; *** *p* < 0.001.

## Data Availability

Not applicable.

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
