# Peer review of "Adaptation of the Pet Bereavement Questionnaire for European Portuguese Speakers"

_ijerph, 2022, doi:10.3390/ijerph20010534_

Round 1
Reviewer 1 Report
Please try to write more concisely (see all yellow markings and comments in the text). Not always easy to understand or follow. Please rewrite extensivly - check syntax and shorten and clarify sentences.

Author Response
Reviewer 1:
Thank you very much for all your comments and questions. They undoubtedly contributed to improve our paper in rigor, clarity and complexity. We answered to all the comments and questions.
Point 1: Convoluted/ shorten and be concise/ syntax/ unclear/ not understandable/? / Sentence to long/ unclear meaning
Response 1: Thank you for your comments. We rewrote extensively the paper, checking syntax and shortening and clarifying sentences, valuing all the yellow markings and comments in the text in order to make it more easily to understand or follow. We also reviewed citation style.
Point 2: Is not the pat a loved one? Rephrase.
Response 2: Thank you for your question/comment. We rephrased the text, clarifying, when we are referring to a pet “loved one” or a human “loved one” (lines 48-50).
Point 3: What ways? Specify.
Response 3: Thank you for your question/comment. In lines 48-56, we clarified the idea that “Research, has long suggested that the loss or death of a beloved pet creates a grief reaction that is, in many ways, comparable to that of the loss of a family member”, including detailed information:
One may wonder if grief over pet death differs from grief over the death of a human loved one. Research has long suggested that the loss or death of a beloved pet creates a grief reaction comparable to that of the loss of a human family member [3]. Some owners described the relationship with their pets as fulfilling an intimate social human role (such as best friend or child), anthropomorphizing them [4]. Hence, the death of a pet could be painful and cause reactions such as anxiety, stress, shame, ambiguous grief, complicated grief and even trauma [5]. These reactions could lead to preoccupation with thoughts of the pet, poor concentration, avoidance behaviour, loss of identity and self-reproach [6]. Bereaved pet owners also reported disruption in daily living and social activities [7].
Point 4: Why veterinarians? Are they qualified?
Response 4: Thank you for your questions. Answering to the questions “Why veterinarians?” and “Are they qualified?”, we included in the text (lines 79-83):
Thus, mental health professionals should be aware of what Hughes et al. [5] and Hess-Holden, et al. [10] referred to as "unprivileged grief" and provide support to bereaved pet owners. Veterinarians, who deal very closely with suffering owners, must also be sensitive to owners’ psychosocial reactions and be capable of directing them to specialized support if needed.
Point 5: how/ when/where/ by whom
Response 5: Thank you for your questions. We answered to the questions how/ when/where/ by whom related to the phrase “However, these factors may not be enough and psychological support should be improved so as to help bereaved owners cope better with grief” (lines 91-93), replacing the sentence by this one:
However, these factors may prove insufficient [13], in which case it is important that veterinarians are prepared to offer psychological first aid and to guide owners towards psychological appropriate professional help where required.
Point 6: Why veterinarians now? It is not their pet or bereavement? Subject matter?
Response 6: We are grateful for the comments on “Why veterinarians now?” “It is not their pet or bereavement”. We eliminated the no essential information and rewrote the phrase (lines 94-97):
Dow, et al. [14] pointed out that most veterinarians did not refer any grieving clients to a psychologist or other mental health professional. Dow et al. [14] further noted that a significant proportion of veterinarians felt that their own mental health was affected when dealing with clients suffering from the loss of a pet.
Point 7: Reference.
Response 7: Thank you for your comment. In line 196, we included the reference of the original version of PBQ: [15].
Point 8: studies show also higher levels of depression with increasing age
Response 8: Thank you for your comment. Concerning the comment “studies show also higher levels of depression with increasing age”, we clarified (lines 429-431):
Moreover, the prevalence of extreme depressive symptoms may decline with age, even though the mean number of depressive symptoms in the population may increase [33].
Please see the attachment.

Reviewer 2 Report
The manuscript “Adaptation of the Pet Bereavement Questionnaire to European Portuguese” is a well-written research paper. The authors draw attention to the issues of the importance of the human-pet relationship, which is of great value to many people. Pets are more and more often perceived as members of family, hence the subject of mourning after the loss of a pet is very important. In many countries, people who mourn the loss of a beloved pet are often ridiculed and their feelings ignored by others. As the authors point out in the Introduction, the role of the veterinarian and his sensitivity in contacts with the owner of the departing animal are very important here. Apart from minor errors, the work is done reliably, the created version in Portuguese adapted to Portuguese realities is well verified.
Comments to the manuscript:
Line 37 - citing the source: "... Podrazik, Shackford, Becker, and Heckert [1]..." - authorship of the article above 2 authors are quoted from “Podrazik et al. [1]” - and so the authors did in line 48. The same applies to lines: 40, 66, 90 - please standardize throughout the text.
Line 56 - explain and expand the DSM-5 abbreviation on the first use
Line 139 - Participants are a bit vaguely described. First, there is information about the participation of 89.3% of women, then after the decimal point –information about 74.6% living in urban areas and age limits of respondents - it seems this information applies only to women and not to all respondents – please describe more clearly.
The manuscript could be accepted for publication in the journal Animals after minor revision.
Author Response
Reviewer 2
Point 1: The manuscript “Adaptation of the Pet Bereavement Questionnaire to European Portuguese” is a well-written research paper. The authors draw attention to the issues of the importance of the human-pet relationship, which is of great value to many people. Pets are more and more often perceived as members of family, hence the subject of mourning after the loss of a pet is very important. In many countries, people who mourn the loss of a beloved pet are often ridiculed and their feelings ignored by others. As the authors point out in the Introduction, the role of the veterinarian and his sensitivity in contacts with the owner of the departing animal are very important here. Apart from minor errors, the work is done reliably, the created version in Portuguese adapted to Portuguese realities is well verified.
Response 1: We would like to thank you for your kind comments on the paper.
Point 2: Line 37 - citing the source: "... Podrazik, Shackford, Becker, and Heckert [1]..." - authorship of the article above 2 authors are quoted from “Podrazik et al. [1]” - and so the authors did in line 48. The same applies to lines: 40, 66, 90 - please standardize throughout the text.
Response 2: Thank you for your comment. References were standardized according the journal´s citation style.
Point 3: Line 56- explain and expand the DSM-5 abbreviation on the first use:
Response 3: Thank you for your comment. We explained and expanded the DSM-5 (Line 60):
“He concluded that pet grief symptoms are consistent with the symptom classification proposed by The Diagnostic and Statistical Manual of Mental Disorders, Fifth Edition (DSM-5) and that they occur in both men and women.”
Point 4: Line 139 - Participants are a bit vaguely described. First, there is information about the participation of 89.3% of women, then after the decimal point –information about 74.6% living in urban areas and age limits of respondents - it seems this information applies only to women and not to all respondents – please describe more clearly.
Response 4: Thank you for your comment. We rewrote participants description in order to be precise and clearer concerning this information (lines 138-145):
“A non-probabilistic convenience sample was studied, consisting of 169 participants who had a pet that died: 89.3% (n=151) female; 89.3% (n=151) living in urban areas; aged between 18 and 82 years (M=34.72; SD=14.07); with a level of education ranging from years 7-9 of basic education to doctorate, with 78.1% (n=132) attending higher education level. All participants, at the time of their participation in the study, had at least one pet, with this number ranging between 1 and 25, and on average they had 3 pets (SD=3.09): 68.6% (n=116) dog, 50.9% (n=86) cat, 7.1% bird (n=12), 10.1% fish (n=17), 6.5% (n=11) guinea pig, 3% (n=5) reptile, and 9.5% (n=16) other animals.
Please see the attachment.

Reviewer 3 Report
Great paper and very well written - enjoyable read!
I only have a couple of queries:
1. Inclusion criteria: to participate an individual needed to have experienced the loss of a pet. Could you provide a little more detail around this - as an eg did this death need to occur within a specified time frame? Was any detail of the death asked about (time frame, natural causes vs accident etc)? Given the subscales of the PBQ it would seem logical that the loss of a pet due to another's actions might relate to higher scores on the anger subscale? What phrasing around eligibility criteria was used as this may introduce a source of bias
2. Multiple analyses were undertaken in the cross validation section - what controls were implemented for the inherent elevation in error rate?
3. The authors assert that the scores on PBQ run counter to what is normally seen in rural populations (object vs subject value of animals) .. however there is evidence that the 'value' or role of an animal can depend on the category ascribed (see pet/pest/profit papers) so a rural individual may well grieve their pet but still be very instrumental in approach to livestock.
Author Response
Reviewer 3
Point 1: Great paper and very well written - enjoyable read!
Response 1: Thank you very much for your comment!
Point 2: I only have a couple of queries: Inclusion criteria: to participate an individual needed to have experienced the loss of a pet. Could you provide a little more detail around this - as an eg did this death need to occur within a specified time frame? Was any detail of the death asked about (time frame, natural causes vs accident etc)? Given the subscales of the PBQ it would seem logical that the loss of a pet due to another's actions might relate to higher scores on the anger subscale? What phrasing around eligibility criteria was used as this may introduce a source of bias.
Response 2: Thank you for your comment.
- One of the inclusion criteria for participants was “(c) having experienced the loss of a pet (participants answered yes to the question "Have you suffered the loss of one of your pets?”) (Lines 214-216).
- In Discussion, we included as limitations of our study the fact of not having controlled the time frame and the causes of the pet loss (Lines 482-486):
“Other limitations relate to the fact that test-retest over a short period of time was not performed, which would have allowed for the exploration of temporal stability of the instrument, nor were the time elapsed since the death of the animal or the circumstances of the animal's death controlled, which could have allowed other analyses of the sensitivity of the PBQ.”
Point 3: 2. Multiple analyses were undertaken in the cross-validation section - what controls were implemented for the inherent elevation in error rate?
Response 2: Thank you for your question. We included more detailed information in the paper (Lines 224-230):
“A sample size must be determined to achieve adequate questionnaire precision and to have enough power to reject a false null hypothesis. A literature review by Anthoine et al. [22] concluded that recommendations range from 2 to 20 subjects per item, with an absolute minimum of 100 to 250 subjects. In the present study the following criteria were defined a priori to assure precision: assessing at least 10 participants for each scale item, i.e., the ratio of respondents to items of 10:1. Furthermore, as there were no missing answers to the items, appropriate handling of incomplete outcome data was not necessary.”
Point 4: The authors assert that the scores on PBQ run counter to what is normally seen in rural populations (object vs subject value of animals) however there is evidence that the 'value' or role of an animal can depend on the category ascribed (see pet/pest/profit papers) so a rural individual may well grieve their pet but still be very instrumental in approach to livestock.
Response 4: Thank you for your comment. In lines 446-453, we try to emphasize this point of view:
“These results are an important contribution towards deconstructing a false preconception among the general population that, in rural areas, animals are still seen essentially as an object and not as a subject (as these concepts are understood by Doré et al. [2]), thus, people living in rural areas would be less likely to develop grief processes in reaction to the death of their animals, a prejudice might lead the community in general and health professionals in particular to underestimate reactions to the death of a pet in people from these areas. Our results suggest that in rural areas there is a clear differentiation between pets, livestock or pest.”
Please see the attachment.

Round 2
Reviewer 1 Report
The subject is interesting and important. In the current document are still a few sentences to clarify or syntax Problems, but these are minor

Author Response
We are grateful for all the comments.
We answered to all comments and questions, in order to improve paper understanding – sentences were clarified and syntax was corrected.
Thank you for your attention.
